# Adsorption of *N*,*N*,*N*′,*N*′-Tetraoctyl Diglycolamide on Hypercrosslinked Polysterene from a Supercritical Carbon Dioxide Medium

**DOI:** 10.3390/molecules27010031

**Published:** 2021-12-22

**Authors:** Mikhail Kostenko, Olga Parenago

**Affiliations:** 1Kurnakov Institute of General and Inorganic Chemistry, Russian Academy of Sciences, Leninsky Prospect 31, 119071 Moscow, Russia; oparenago@scf-tp.ru; 2Chemistry Department, Moscow State University, Leninskie Gory 1, Bldg. 3, 119234 Moscow, Russia

**Keywords:** adsorption, *N*,*N*,*N*′,*N*′-tetraoctyl diglycolamide, hypercrosslinked polystyrene, supercritical fluid, chromatography, gravimetry, isotherm

## Abstract

The work considers for the first time the preparation of sorbents based on hypercrosslinked polysterene (HCP) and chelating agent *N*,*N*,*N*′,*N*′-tetraoctyl diglycolamide (TODGA) by impregnation in the supercritical (SC) CO_2_ medium. Such sorbents can be applied for further isolation and separation of lanthanides, actinides and other metals. They are usually prepared by impregnation in toxic organic solvents (e.g., methanol, dichloromethane). Our study shows that application of SC CO_2_ instead of organic solvents can significantly speed up the impregnation, perfom it in one stage and make the process more eco-friendly. At the same time, the obtained sorbents are close in their parameters to the classical ones. This article presents the results of measuring the TODGA adsorption isotherms on two HCP sorbents (MN202 and MN270) on a wide range of SC fluid parameters. Adsorption measurements were carried out using on-line supercritical fluid chromatography and gravimetry. Based on the sorption capacity parameter, MN202 sorbent was selected as the better carrier for TODGA. An impregnation temperature increase within the range 313–343 K in isochoric conditions (ρ = 0.780 g/mL) reduces the maximum of TODGA adsorption from ~0.68 mmol/g to ~0.49 mmol/g.

## 1. Introduction

Metal extraction from aqueous solutions is a common task both in laboratory practice and in industrial production. This problem is solved quite successfully by using chelating agents. Quite promising among them are diamides, and, in particular, *N*,*N*,*N*′,*N*′-tetraoctyl diglycolamide (TODGA), which can be used for extraction of a variety of rare earth elements and radioactive metals of the actinide family [1,2,3]. That is why one of the potential applications of TODGA is purification of waste water in the nuclear industry, removing radionuclides [2,4,5]. TODGA is normally used as a solution in aliphatic hydrocarbons [3,4,5] but for many problems it is more convenient to use adsorbents impregnated by TODGA. Such materials can adsorb metals from acidic aqueous solutions, which allows them to be used in solid phase extraction and extraction of metals from complex mixtures. Previous research [1,6,7,8,9,10,11,12] shows that it is possible to use impregnated sorbents based on TODGA and other similar ligands for chromatographic separation of mixtures containing lanthanides, actinides and other metals of valences II, III and IV. The role of carrier matrices in production of sorbents of this type can be played by materials of different kinds, such as silica gels modified by polymers [7], graphene aerogels [13], hypercrosslinked polystyrene (HCP) [1,14], etc. Such carriers are commonly impregnated with TODGA solutions in volatile solvents (methanol, dichloromethane, etc.). The solvents are removed by evaporation (reaching complete deposition of the chelating agent on the carrier) [1,6,8,10,11,12] or the carrier with a certain amount of the adsorbed chelating agent is filtered out of the solution residue [7,13]. Such approaches to impregnation require volatile toxic organic solvents and, in addition, are rather labor- and time-consuming, which encourages us to use more environmentally friendly and cheaper solvents such as supercritical (SC) CO_2_.

In this work, we applied two variants of commercially available HCP as the TODGA carriers. The unique structure of this class of materials provides them with a number of features, such as high porosity, stability within a wide range of conditions and a surface with adsorptive aromatic centers. This allows HCP to act as an adsorbent in a number of industrial, medical and scientific applications [15]. The use of adsorbents of this type in SC fluid media has not been sufficiently studied yet and is of special scientific interest.

The methods of adsorption measurement of SC fluid solution components are largely analogous to those used to measure adsorption from liquid solutions. The methods can be divided into dynamic and static. The dynamic adsorption measurement methods are based on analyzing the substance penetration through an adsorbent layer in a column. There are examples of successful applications of dynamic methods (column breakthrough [16,17,18] and chromatographic [19,20,21,22,23,24,25] methods) to measure adsorption in SC fluid media. However, the accuracy of the dynamic methods is largely dependent on the column packing with an adsorbent due to the hydrodynamic effects [26]. This means that these methods are not reliable in case of HCP studied in our work as we have earlier determined in our laboratory that the use of such materials in SCF leads to changes in the particle size caused by swelling and destruction of the column packing layer [27]. 

The principle underlying static adsorption measurement methods is reaching the equilibrium of substance distribution between the bulk phase and the adsorbed layer. When measuring adsorption from the liquid phase, an adsorbent sample is introduced into a solution with a pre-known concentration of the adsorbed substance, after which an equilibrium is established in the system at the required temperature. The amount of the adsorbed substance is determined based on the system mass balance [26]. Adsorption measurement from the SCF phase is complicated because the bulk phase is under pressure. For this reason, the experiment is usually conducted by a different technique: after reaching the adsorption equilibrium, the bulk phase is removed from the system and the substance content on the adsorbent surface is determined by a variety of methods (such as gravimetry or desorption into the liquid phase with subsequent spectrophotometric or chromatographic analysis [28,29,30]). Such approaches are relatively easy to implement but, as is known [28], a pressure relief makes SCF lose their dissolving capacity, which may lead to substance deposition in the sorbent pores. This, in turn, may lead to errors in adsorption measurement. We have earlier [31] proposed a static approach to sorption analysis in SC media based on on-line supercritical fluid chromatography (SFC). A similar approach was applied in several works to determine the substance solubility in SCF [32,33,34,35] and distribution coefficients in the liquid-SCF system [36,37,38,39,40], and to control chemical reactions [41,42]. It allows fast quantitative analysis of the considered system components without depressurization or special sample preparation.

In this work, we consider for the first time the preparation of sorbents based on HCP and chelating agent (TODGA) by impregnation in the SC CO_2_ medium. Such sorbents can be applied for further isolation and separation of metals. The main purpose of the work was to study the adsorption of TODGA on HCP in the SC CO_2_. This is necessary for the development of more eco-friendly methods for producing sorbents based on TODGA and other chelating agents. In the paper, measurements of the TODGA adsorption isotherms on two variants of HCP are presented, the effect of the SC fluid density and temperature on adsorption is estimated, and a brief comparison of the procedure for sorbents preparation in methanol and SC CO_2_ media is performed. Besides, special attention is paid by the authors to optimization and comparison of adsorption measurement methods in the considered conditions. The on-line SFC method has been used to measure adsorption only once before so it is relatively new. For this reason, its advantages over the gravimetric method are considered in more detail in the framework of the study.

## 2. Results and Discussion

### 2.1. The Experimental Unit Testing and Calibration 

Since the experimental unit described in [31] had been significantly modified, before making the main measurements, we conducted preliminary tests to check the stability of temperature maintenance and sealing of all the unit components. The tests were made at 313 K, the autoclave was filled with CO_2_ until the pressure reached 20 MPa, after which we monitored the temperature and pressure values for 3 days. The maximum deviation of the temperature value in the autoclave from the pre-set value over the whole period was ±0.4 K. The pressure decrease after 72 h was about 0.15 MPa, which, by the NIST Chemistry WebBook data [43], corresponds to CO_2_ leakage in the amount of about 0.20 g (the calculated initial mass was 121.77 g). We considered such results satisfactory enough to proceed with the main experiments.

To calculate the concentrations of TODGA in the autoclave, we had to measure its effective volume. The geometric volume of the autoclave was 150 mL but it also contained the magnetic stir bar, the vial and the support. The autoclave effective volume was measured by filling it step by step with distilled water using 100–1000 µL and 1–10 mL Thermo Scientific Lite (Lenpipet Thermo Scientific, Moscow, Russia) calibrated mechanical pipettes. The measurements were made three times. Taking into account the pipette error, the effective volume of the autoclave was 145 ± 1 mL.

The TODGA retention time in the selected analysis conditions was 1.13 ± 0.02 min. The calibration tests showed high reproducibility of the results obtained in the considered experimental unit. In all the tests, the sample from the autoclave was injected at least twice (Figure 1), to exclude accidental error. Since the TODGA concentration in the autoclave decreased after the sample collection, the area of the next peak was always 0.3–0.5% smaller than that of the previous one. Taking this into account, we were able to estimate the approximate rate of TODGA sample dissolution in the experimental conditions. It was established that increasing the dissolution time from 10 to 60 min did not increase the chromatographic response. This indicates that the TODGA samples completely dissolved in less than 10 min.

By analyzing a number of TODGA samples, we plotted a calibration dependence of the amount of the substance introduced into the autoclave on the chromatographic peak area (Figure 2). The dependence equation takes the form: (1)$$n=Resp\cdot 7.8804\cdot 10^{-8}$$
where Resp is the chromatographic response (peak area).

Most of the calibration points were obtained at a pressure of 20 MPa. However, to confirm its performance in the extended CO_2_ density range, we obtained several points in the pressure range from 10 to 30 MPa. When the pressure in the system was reduced to 10 MPa, there were nonreproducible errors in the analyzed points, which are characteristic of conditions of incomplete substance dissolution. This effect was observed even with relatively small concentrations of TODGA. For this reason, the points obtained at 10 MPa were not used to construct calibration dependence.

The calibration experiments confirmed the high accuracy of the measurements made on the modified experimental unit. The maximum absolute deviation of the experimental value ($n_{i}$) from the one calculated by calibration ($\hat{n_{i}}$) was 2.2 µmol, the determination coefficient: *R*^2^ = 0.9998. The root mean square error (RMSE) of approximation was calculated as follows:(2)$$\mathrm{RMSE}=\sqrt{\frac{\sum_{i}{\left(n_{i}-\hat{n_{i}}\right)}^{2}}{N}}$$
where *N* is the number of experimental points. The RMSE value was 0.0012 mmol.

### 2.2. TODGA Adsorption on MN202 and MN270 from a Solution in SC CO_2_

MN202 is known to be applicable as a TODGA carrier. For example, it is used as the basis for producing BAU-1M, a commercial sorbent (Sorbent-Tekhnologii, Moscow, Russia) [1,14]. In addition to MN202, we used the MN270 sorbent with smaller pores and a larger specific surface area. It was interesting for us to check whether it could be applied to solve the problem under consideration. We therefore plotted isotherms of TODGA adsorption on both HCP variants at 313 K and medium pressure of 20 MPa (Figure 3).

In the first experiments, we estimated the approximate time required for the system to reach the adsorption equilibrium. It was established that as time increased in the sequence of 30, 60 and 120 min, the TODGA content in CO_2_ did not change significantly and, consequently, the equilibrium in these conditions was reached in less than 30 min. Nevertheless, to make sure the obtained results were correct, the impregnation time in all the other experiments was 60 min. 

The adsorption isotherms are described quite well by the Langmuir model:(3)$$q=q_{s}\frac{K\cdot C}{1+K\cdot C}$$
where q is the adsorption, $q_{s}$ is the adsorbed monolayer capacity, K is the adsorption equilibrium constant, C is the TODGA concentration in the bulk phase. 

Overall comparison of Langmuir model parameters of TODGA adsorption isotherms is given in Table 1. In the considered conditions, the saturation of the TODGA monolayer on the MN202 surface was reached at approximately 0.64 mmol/g; the mass equivalent of this value, 0.37 g/g, is close to the values of the analogous commercially available sorbents’ load [1]. Interestingly, the maximum adsorption on MN270 is much lower, at 0.10 mmol/g. Since the nature of the sorbent surface is identical, this is most probably caused by the difference in the pore size and accessibility. The sorbents’ declared characteristics [44,45] show that the only significant difference of MN270 from MN202 is the higher degree of crosslinking, which reduces the mesopores’ average size from 220 Å to 80 Å and increases the specific surface area. The specific surface area of the sorbents in a non-swollen state measured by low temperature nitrogen adsorption was 590 ± 40 m^3^/g and 980 ± 50 m^3^/g for MN202 and MN270, respectively. 

The isotherms show a wide dispersion of experimental adsorption value points, sometimes much higher than the calculated absolute measurement error. This is assumed to be the result of the difference between the sizes of the adsorbent particles (the sorbent particle diameter is from 0.3 to 1.2 mm) and pore accessibility. In the experiments, we used relatively small sorbent samples (about 0.05–0.15 g), which could lead to a considerable spread of the sample specific surface area values in different experiments as the sorbents were somewhat inhomogeneous.

The MN202 sorbent turned out to be a better carrier for TODGA and was used in further experiments.

### 2.3. Comparison of the Adsorption Measurement Approaches

The values obtained by measuring the adsorption gravimetrically at the initial part of the isotherm are very close to those obtained by chromatographic analysis (Figure 3). As the TODGA concentration in the solution increases, the results of the gravimetric measurements become overestimated. This is the result of the higher intensity of TODGA deposition from a CO_2_ solution to the HCP surface after depressurization. This effect leads to increase of the sorbent mass, regardless of the equilibrium-adsorbed TODGA. Some of the adsorption values measured gravimetrically in this work were more than 25% higher than those obtained by chromatography. Presumably, the accuracy of the method can be improved by wrapping the sorbent with a porous material that would reduce the available volume, from which the substance can precipitate on the sorbent under the depressurization. However, the layer of shielding material will reduce the rate of the mass transfer between the sorbent surface and the solution phase, which may considerably increase the time required for the system to reach the equilibrium state and nullify one of the main advantages of the experimental unit—the high measurement speed. Moreover, the material itself can potentially act as an adsorbent and cause error in the adsorption measurement, especially if the adsorption capacity of the main adsorbent is not high. Filter paper is often used to shield sorbents [28,29,30]. Cellulose, which is the main component of filter paper, is known to be an active adsorbent for many classes of chemical compounds [46,47,48] and is used as a sorbent (stationary phase) in paper chromatography.

In the present work, gravimetry was used as an additional control method. For this reason, we did not shield the sorbent with any porous materials, in order to minimize the error of the adsorption measurement by the chromatographic method and to minimize the time required for the system to reach the adsorption equilibrium. To limit TODGA deposition on the adsorbent after depressurization, we covered the upper part of the vial with perforated aluminum foil. This version of the gravimetric method had satisfactory results at the initial section of the isotherms, which confirmed that the obtained data were correct.

Thus, on-line SFC analysis proved to be a quick and accurate method for adsorption measurement in this work. The gravimetric method is easier to implement experimentally but it should be used with caution, especially when measuring adsorption from concentrated solutions.

### 2.4. Fluid Density Effect on TODGA Adsorption

In this work, we found out that change in the fluid medium density by pressure variation in the range of 15 to 30 MPa has no significant effect on TODGA adsorption on MN202 (Figure 4). 

This fact is probably associated with the relatively low energy of CO_2_-HCP interaction compared to the energy of the sorbent surface interaction with TODGA. For this reason, changes in the density and, hence, amount of CO_2_ in the considered system do not produce a significant effect on the TODGA–HCP adsorption equilibrium. This, in turn, makes it possible to produce impregnated sorbents at relatively low pressure, which is especially important when moving from laboratory conditions to industrial production.

### 2.5. Temperature Effect on TODGA Adsorption

Deviations from the Langmuir model were observed during temperature variation in the range 313–343 K under isochoric conditions (ρ = 0.780 g/mL) (Figure 5). 

According to the Langmuir model, temperature does not affect the monolayer capacity but can only change the adsorption equilibrium constant influencing the isotherm curvature. However, there was a noticeable reduction in the monolayer capacity when the temperature went up, which was due to the peculiarities of adsorption thermodynamics. Since adsorption is an exothermic process, the temperature increase reduces the maximum equilibrium concentration of an adsorbate on the sorbent surface. For a general description of adsorption, taking into account the influence of temperature, semi-empirical models based on the Langmuir equation are often used [49,50,51]. Various temperature functions are used in these models instead of the monolayer capacity fixed value, and the equilibrium constant dependence on temperature is described based on its thermodynamic meaning:(4)$$K=K_{0}\cdot exp\left(\frac{E}{R\cdot T}\right)$$
where *K*_0_ is the reference constant (entropy multiplier), *E* is the heat of adsorption.

In this paper, we used the following model equations to describe the adsorption dependence on concentration and temperature:(5)$$q=\left(a-b\cdot T\right)\frac{K_{0}\cdot exp\left(\frac{E}{R\cdot T}\right)\cdot C}{1+K_{0}\cdot exp\left(\frac{E}{R\cdot T}\right)\cdot C}$$
(6)$$q=\left(a\cdot exp\left(b\cdot T\right)\right)\frac{K_{0}\cdot exp\left(\frac{E}{R\cdot T}\right)\cdot C}{1+K_{0}\cdot exp\left(\frac{E}{R\cdot T}\right)\cdot C}$$
where *a*, *b* are the empirical coefficients.

Both model equations describe experimental results quite well. The parameters of Equations (7) and (8) obtained by approximation are given in Table 2.

There is thus no reason to raise the impregnation temperature when preparing the target adsorbent, which is convenient in terms of process implementation.

### 2.6. TODGA Adsorption on MN202 from Methanol Solution

In this study, we also built an isotherm of TODGA adsorption on MN202 from a methanol solution at 313 K (Figure 6) by the classical analysis of liquid phase above an adsorbent. In such conditions, the Langmuir model is unsuitable for describing experimental data. This fact is assumed to be associated with significant adsorption ability of the HCP surface towards methanol [52]. Besides, methanol can effectively solvate the TODGA molecule, which follows from complete mutual dissolution of these compounds in the conditions considered in this work. Thus, in the HCP/TODGA/methanol system, more complex interactions can appear in comparison with the HCP/TODGA/CO_2_ system, associated with the competition between TODGA and methanol for active adsorption sites on the HCP surface. It should be also taken into account that the degree of swelling of HCP varies in different solvents, which may also lead to some differences in the HCP adsorption properties in methanol or CO_2_ media [53,54].

The Langmuir model is among the simplest ones. It describes in theoretical terms the ideal homogeneous adsorption of gases (adsorption on the surface with a regular arrangement of adsorption centers, identical in energy, with no adsorbate-adsorbate interactions). It is often suitable for describing adsorption from liquid solutions but, in such cases, it should be considered as a semi-empirical model because it contradicts the model postulate about the absence of interaction between the adsorbate particles and between the adsorbate and solvent particles, which leads to misinterpretations of the thermodynamic meaning of the adsorption equilibrium constant [55]. There are numerous empirical and semi-empirical models that take into account adsorption heterogeneity and other factors to describe complex isotherms. In the considered case of TODGA adsorption from a solution in methanol, the bi-Langmuir and Freundlich models demonstrated reasonably good approximations. The parameters of the model equations are given in Table 3.

Unfortunately, the wide points dispersion on the isotherm does not allow us to identify the details of the adsorption mechanism based on the data obtained. It can be cautiously assumed that the applicability of the bi-Langmuir model indicates that there are two or more independent types of adsorption centers on the HCP surface that could be the result of, for example, sorbent swelling or reversible modification of its surface with solvent molecules.

Figure 6 shows that at a TODGA concentration in the methanol solution of about 7 mmol/l, the adsorption value is only 0.20 ± 0.03 mmol/g, with the adsorbent remaining unsaturated. In case of adsorption from CO_2_, the isotherm plateau is reached already at 1–1.5 mmol/l. It is thus not advantageous to prepare impregnated TODGA adsorbents from methanol in equilibrium adsorption conditions. Such adsorbents are mainly prepared by complete joint evaporation of the TODGA solution in a volatile solvent in the presence of a carrier matrix. Preparation of impregnated sorbents from a solution in SC CO_2_ is characterized by a higher process speed and application of a cheap and nontoxic solvent. The increased diffusion coefficients in the SC medium and the absence of interfacial tension allow the substance to be quickly delivered to the pores of the adsorbent. After the impregnation is completed, CO_2_ can be easily removed from the autoclave in the gaseous state, and the obtained product does not require additional treatment, such as drying from a solvent. Of course, it must be taken into account that switching to industrial scale requires solving a number of problems related to the transition to a continuous or half-periodic variant of the process and design of the corresponding experimental unit and technique. Nevertheless, the obtained results confirm the good prospects of the work in this field.

### 2.7. Comparison of Stability of Impregnated TODGA Adsorbents

Since TODGA is used for metal adsorption from acidic solutions, in this work we compared the resistance of impregnated adsorbents to being washed out by nitric acid. We prepared samples through equilibrium adsorption from SC CO_2_ (sample 1) or evaporation from methanol solution (sample 2) with the TODGA content of 0.28 g/g and 0.31 g/g, respectively, according to the method described in the experimental section. After washing out the adsorbents with water and nitric acid solutions, we calculated the decrease in TODGA concentration in the adsorbent samples (Figure 7). The water did not have a leaching effect on impregnated adsorbents; however, as it was expected, in acidic solutions the formation of TODGA complexes with nitric acid [56] led to the extraction of the chelating agent into the solution. The leakage of TODGA and other ligands is a common feature of such sorbents reported in many works [6,12,14]. Despite this fact, they can be successfully applied in various separation processes in practice.

The resistance of adsorbent samples 1 and 2 to being washed out with water and nitric acid solutions was almost identical, taking into account experimental error and an initially higher TODGA content in sample 2. This indicates that the impregnation method does not significantly affect the described property of the obtained product. 

## 3. Materials and Methods

### 3.1. Materials

The MN202 and MN270 (Purolite, Llantrisant, UK) sorbents were provided by S. Lyubimov (INEOS RAS, Moscow, Russia). Before use, the sorbents were carefully washed with acetone (99.85 wt%, Khimiya XXI Vek, Moscow, Russia) for 2 h at room temperature, with the sorbent/acetone mass ratio approximately equal to 1/10. After the washing, the sorbents were filtered on paper filters and dried in a drying oven at 393 K for 24 h. Then, the sorbent samples were kept in a desiccator in the presence of P_2_O_5_ (98 wt%, Khimiya XXI Vek, Moscow, Russia) as the drying agent. Physical and chemical characteristics of sorbents are presented by the manufacturer on the official website [44,45].

TODGA (Figure 8) (≥98%, JSC “Axion—Rare and Precious Metals”, Perm, Russia) was not additionally purified before usage. 

Methanol (extra pure, Chimmed, Moscow, Russia) was used as the cosolvent in the supercritical fluid chromatography, TODGA impregnation to HCP and re-extraction of the sorbents.

Distilled water obtained using a ListonA1104 distiller (Liston, Zhukov, Russia) and nitric acid (≥64 wt%, Khimiya XXI Vek, Moscow, Russia) were employed to evaluate the stability of impregnated adsorbents in an acidic aqueous medium.

### 3.2. Equipment

The adsorption measurements in the work were made in a modified version of the experimental unit assembled earlier for direct chromatographic analysis of solutions in SC fluids [31] (Figure 9). Namely, we used a 150 mL autoclave designed for void volume minimization. A C-MAG HS 7 magnetic stirrer (IKA, Staufen, Germany) was employed to mix the medium in an autoclave. Thermostatic control utilised an electric heating jacket and based on the data from a thermocouple placed inside the autoclave, with TRM202 (Owen, Moscow, Russia) as the controller. The thermocouple and the controller were calibrated by the readings of a tested liquid thermostat, with the deviation of the readings within the range 298–348 K not exceeding 0.5 K. All the sampling lines were minimized in length and volume, and temperature was controlled using a liquid thermostat consisting of a submersible M02 unit (Termex, Tomsk, Russia). The pressure in the system was measured by an APZ-3420 electronic transducer (Piezus, Moscow, Russia) with the maximum absolute error of ±0.1 MPa.

The principle of the unit operation consists in sampling a certain amount of the medium from the autoclave, where the process under study is realized, into the chromatograph sample loop for subsequent analysis. The volume of the sample loop in all the experiments was 10 µL. The sample collection into the loop was made by expanding the volume of the medium under pressure and allowing its flow into a special capillary with a needle valves on each end (sampling device). The capillary volume was 250 µL. Supercritical fluid chromatography enables a sample to be analyzed directly, without depressurization or additional sample preparation, which reduces the possibility of introducing additional error.

A Waters Acquity UPC^2^ chromatograph (Waters, Milford, MA, USA) was used as the analytical instrument in the experimental unit. The chromatograph consisted of:CO_2_ and cosolvent pump (Acquity ccBSM);Acquity UPLC autosampler;column thermostat (Acquity Column Manager);diode-array detector (Acquity UPC^2^ PDA);flow control unit and a back pressure regulator (Acquity Convergence Manager).

The sorbent and TODGA samples were weighed on an Ohaus Pioneer PX225D semi-microbalance (Ohaus, Parsippany, NJ, USA).

The adsorbent specific surface area values were determined by the low temperature nitrogen adsorption method using an ATX-06 sorption unit (Katakon, Novosibirsk, Russia) by the BET model. Five points were measured within the range of nitrogen partial pressure values 0.05–0.25. The removal of the adsorbed moisture from the samples before the surface area measurement was carried out in a dry helium flow at 393 K and took 60 min. 

### 3.3. Quantitative Analysis of TODGA

The TODGA samples were analyzed by the SFC method in the following conditions: column—Luna C18-2 (150 × 4.6 mm, 5 µm, Phenomenex, Torrance, CA, USA), column temperature—308 K, mobile phase flow rate—3 mL/min., mobile phase composition—CO_2_/methanol (95/5 vol%), back pressure in the system—10.5 MPa. The detection was conducted at the wavelength of 215 nm.

The quantitative determination of the TODGA content in the autoclave was carried out using a calibration dependence. A TODGA sample in a glass vial was placed into the autoclave, the system was thermostatically controlledand CO_2_ was fed by a Supercritical 24 pump (Teledyne SSI, State College, PA, USA) until the target pressure value was reached. The carbon dioxide feeding rate was about 10 mL/min. in liquid state under the pump head cooling regime. The system was intensively stirred until the sample dissolved in SC CO_2_. Sampling from the autoclave was then conducted. To do that, we opened the first valve of the sampling device and let part of the autoclave medium flow under pressure through the sample loop and fill it. By turning the 6-port valve, we injected the sample into the flow of the chromatograph mobile phase, where we performed analysis under the conditions described earlier. We then closed the first valve of the sampling device and opened the second one to remove the sample residue. Before the next sample was injected, we returned the valves to their initial positions. Based on the data obtained, we plotted the calibration curve of the TODGA amount in the autoclave on the chromatographic peak area.

### 3.4. Measurement of TODGA Adsorption from a SC CO_2_ Solution

Two approaches to TODGA adsorption measurement in an SC CO_2_ medium were applied in the work. The first approach had been earlier described by us [31] and consists of direct chromatographic analysis of the solution bulk phase being in a thermodynamic equilibrium with the adsorbent. Based on the system’s material balance and the residual TODGA concentration in CO_2_ determined by the analysis, we calculated the adsorption value by the formula:(7)$$q=\frac{n_{0}^{TODGA}-n^{TODGA}}{m_{0}^{ads}}$$
where $n_{0}^{TODGA}$ is the initial TODGA amount in the solution, $n^{TODGA}$ is the TODGA amount in the solution after adsorption equilibrium is reached, $m_{0}^{ads}$ is the adsorbent mass.

The alternative approach to adsorption measurement used in the work was the gravimetric method [28,30], calculating the amount of the adsorbed substance by measuring the changes in the adsorbent mass during adsorption in static conditions:(8)$$q=\frac{\left(m^{ads}-m_{0}^{ads}\right)}{m^{ads}\cdot M^{TODGA}}$$
where $m^{ads}$ is the adsorbent mass after the experiment, $M^{TODGA}$ is the TODGA molar mass.

Both approaches were realized simultaneously in the same apparatus. Adsorbent and TODGA samples in a glass vial covered with perforated aluminum foil were placed into the autoclave. The autoclave was sealed and the system was thermostatically controlled. CO_2_ was pumped into the experimental unit at the rate of about 10 mL/min under the pump head cooling regime. After reaching the target temperature and pressure values, the system was stirred intensively until thermodynamic equilibrium was reached. We then sampled the fluid phase for TODGA SFC analysis. Next, the autoclave was depressurized, the sorbent was taken out of the autoclave and weighed. This allowed us to make a direct comparison of the results obtained in identical conditions applying the two approaches.

### 3.5. Measurement of TODGA Adsorption from a Methanol Solution

Adsorbent, methanol and TODGA samples weighed on a semi-microbalance were placed into a 40 mL vial, which was sealed, carefully mixed and placed into a liquid thermostat at 313 K for 24 h. Then 2-3 mL of liquid phase samples were collected and filtered through a syringe filter (PTFE with the pore diameter of 0.2 µm). The TODGA concentration in the sample was determined by the SFC method in accordance with the calibration curve prepared in advance. The adsorption was calculated based on the TODGA mass balance between the bulk and adsorbed phases:(9)$$q=\frac{\left(C_{0}^{TODGA}-C^{TODGA}\right)\cdot V_{MeOH}}{m_{0}^{ads}}$$
where $C_{0}^{TODGA}$ and $C^{TODGA}$ are the TODGA concentrations in methanol at the beginning of the experiment and after the adsorption equilibrium is reached, respectively, $V_{MeOH}$ is the methanol volume.

### 3.6. Preparation and Comparison of Impregnated Adsorbents’ Stability

Sorbents based on MN202 impregnated with TODGA were obtained by the following methods:(1)Impregnation from a liquid solvent. Sorbent and TODGA samples (4.00 and 1.32 g, respectively) were placed into a 100 mL round-bottom flask in order to prepare a species with the chelating agent content of about 30 g/g. After that, 20 mL of methanol were added to the mixture under constant stirring. The flask was covered with a lid and stored for 2 h. The methanol was then distilled off by a vacuum rotary evaporator at a temperature of 353 K for 3 h. The target product was taken out of the flask and placed in a weighing bottle, which was then sealed.(2)Impregnation from SC CO_2_. A glass vial with sorbent and TODGA samples was placed into the autoclave. The sample mass values were calculated based on adsorption isotherms plotted in advance to prepare impregnated forms with the TODGA content of about 30 g/g. CO_2_ was then fed into the system under constant stirring to reach the target pressure value of 20 MPa. The medium temperature was 313 K. The system was kept for 1 h, after which chromatographic analysis of the medium above the sorbent was performed. The autoclave was then depressurized, the product was taken out of the autoclave and was placed in a weighing bottle, which was then sealed.

The actual final content of TODGA in both samples was determined gravimetrically as the difference between the sorbents’ masses after and before impregnation. The stability of the obtained sorbents in HNO_3_ aqueous solutions was determined by submerging a sample (0.500 ± 0.001 g) in 10 mL of water or a nitric acid solution with the concentration of 1 or 3 M. The samples were kept at room temperature (295 ± 2 K) for two days under regular stirring.The liquid phase was then poured out, the samples were placed on filter paper and the residual moisture was removed by air-drying. The TODGA content in the samples after the treatment was checked by re-extracting TODGA from HCP by methanol with subsequent chromatographic analysis. Samples of 0.060 ± 0.005 g were used for the re-extraction. Each sample was washed with methanol three times in portions of 15 mL each for 8–12 h, after which all the extracts were placed in a 50 mL volumetric flask and diluted with methanol to scale. Using SFC, we found the TODGA concentration in the obtained solutions and its mass fraction in the impregnated sorbent species after the treatment.

## 4. Conclusions

The present work studied the adsorption of TODGA, a common chelating agent, on HCP in an SC CO_2_ medium. The measurements were made by two methods: on-line SFC and gravimetrical analysis. Both methods showed almost identical results, but the first proved to be more reliable when measuring adsorption from concentrated solutions.

Isotherms of TODGA adsorption on the MN202 and MN270 sorbents were measured at 313 K and 20 MPa. Based on the capacity of the sorbents determined by isotherm approximation, the MN202 sorbent was chosen as a suitable carrier for TODGA.

Varying the fluid density during impregnation within the range 0.78–0.91 g/mL in isothermal conditions did not have a significant effect on TODGA adsorption. In contrast, the temperature parameter has a significant influence on the MN202 TODGA adsorption capacity. Semi-empirical models based on the Langmuir equation were used to make a general description of the adsorption dependence on concentration and temperature. The temperature increase from 313 K to 343 K at the medium density of 0.78 g/mL changed the maximum amount of adsorbed TODGA from ~0.68 mmol/g to ~0.49 mmol/g.

In conditions of equilibrium adsorption from methanol it is inappropriate to prepare impregnated TODGA/HCP adsorbents, due to the small slope value of the isotherm. The SC CO_2_ application in the impregnation procedure allowed us to reduce the process time and eliminate toxic organic solvents. The samples of the sorbent impregnated in methanol and SC CO_2_ have identical chemical resistance to nitric acid solutions, so there is no difference between two impregnation media with respect to this property.

## Figures and Tables

**Figure 1 molecules-27-00031-f001:**
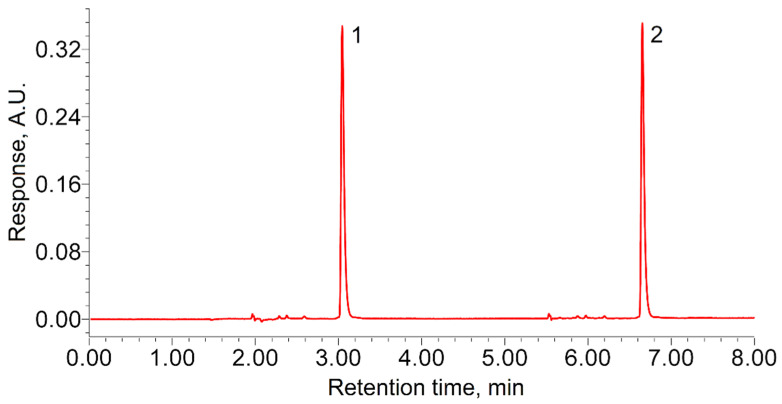
Example of two-time injection of a TODGA sample from the autoclave.

**Figure 2 molecules-27-00031-f002:**
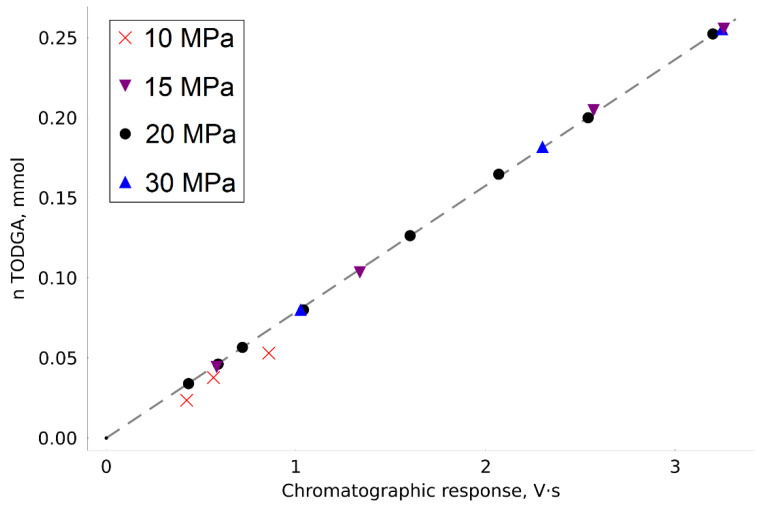
Calibration dependence of TODGA content in the autoclave on the chromatograph response (the dashed line is linear approximation of calibration points).

**Figure 3 molecules-27-00031-f003:**
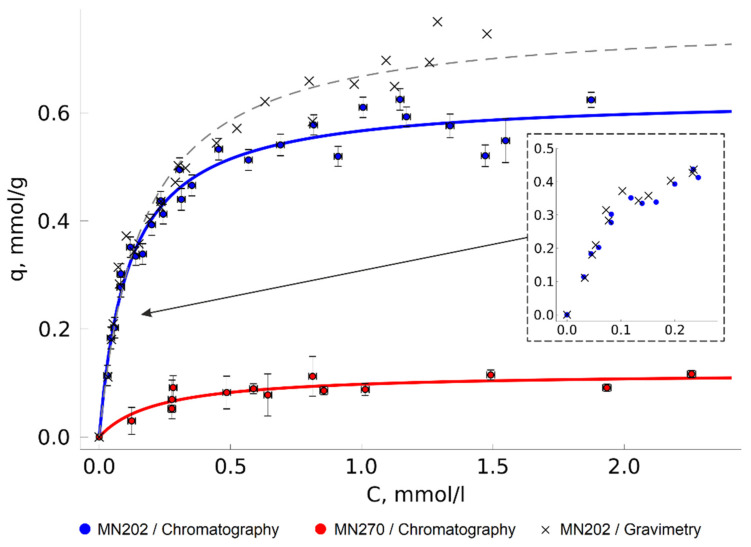
Isotherms of TODGA adsorption on MN202 and MN270 from a solution in SC CO_2_ (T = 313 K, P = 20 MPa).

**Figure 4 molecules-27-00031-f004:**
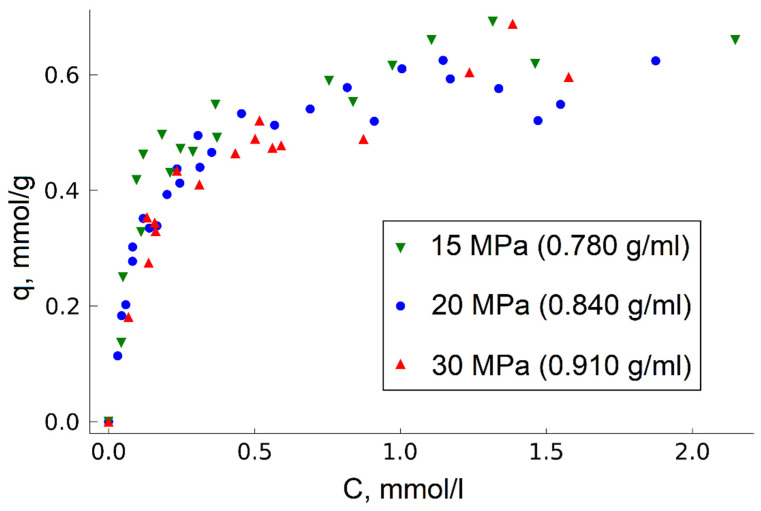
Effect of fluid density on TODGA adsorption (T = 313 K).

**Figure 5 molecules-27-00031-f005:**
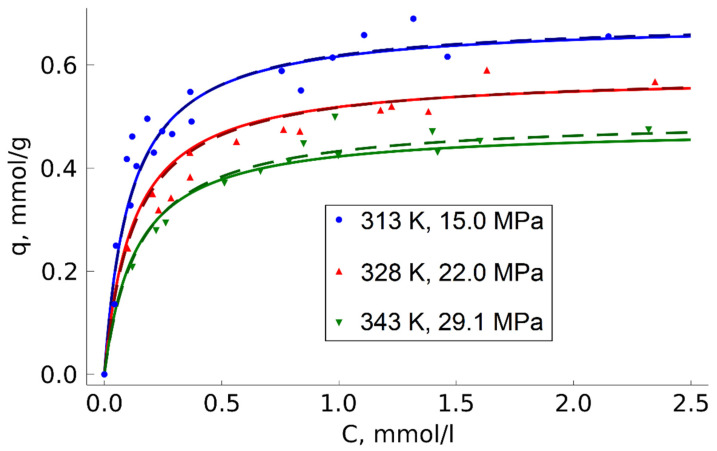
Temperature effect on TODGA adsorption on MN202; approximation of experimental data by Equations (7) (dashed line) and (8) (solid line).

**Figure 6 molecules-27-00031-f006:**
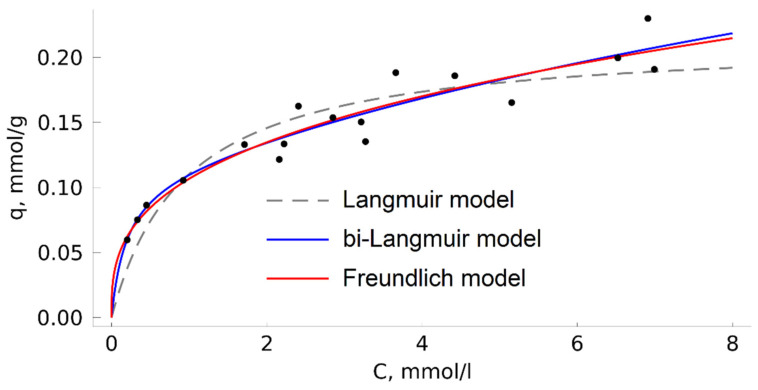
Isotherm of TODGA adsorption on MN202 from a solution in methanol at 313 K.

**Figure 7 molecules-27-00031-f007:**
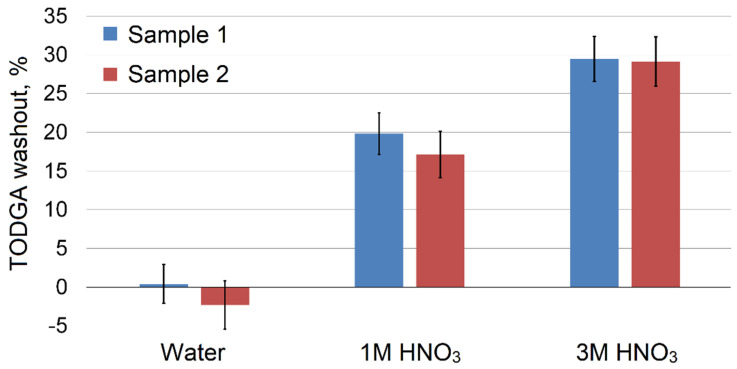
Reduction in TODGA content in sorbents washed with water and nitric acid solutions.

**Figure 8 molecules-27-00031-f008:**
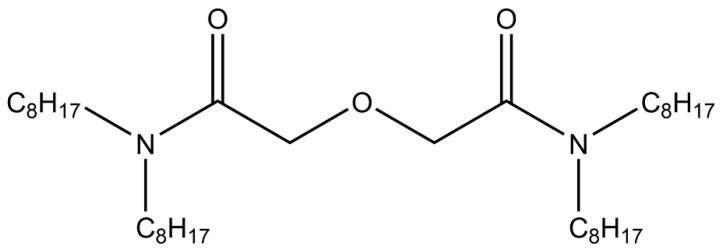
TODGA structural formula.

**Figure 9 molecules-27-00031-f009:**
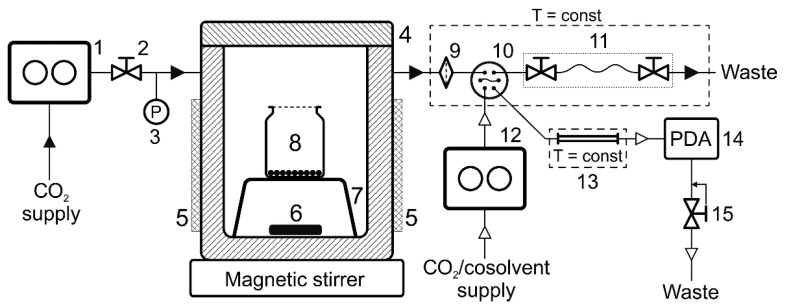
Scheme of the experimental unit: 1—CO_2_ pump, 2—valve, 3—pressure transducer, 4—autoclave, 5—heating jacket, 6—magnetic stir bar, 7—support, 8—glass vial with an adsorbent, 9—in-line filter, 10—6-port valve with a sample loop, 11—sampling device, 12—chromatograph pump, 13—chromatographic column, 14—detector, 15—automatic back pressure regulator.

**Table 1 molecules-27-00031-t001:** The Langmuir model parameters of TODGA adsorption isotherms (T = 313 K, P = 20 MPa).

Adsorbent	$q_{s},\;\mathrm{mmol}/g$	*K*, L/mol	*R* ^2^
MN202	0.64	8.92 · 10^3^	0.9688
MN270	0.10	4.39 · 10^3^	0.8519

**Table 2 molecules-27-00031-t002:** Parameters of the model equations for describing the adsorption temperature dependence.

Equation	a, mol/g	b	$K_{0},\;L/mol$	E, J/mol	$R^{2}$
**(7)**	2.82·10^−3^	6.84·10^−6^ mol/(g·K)	900.24	6030.05	0.9181
**(8)**	20.66·10^−3^	−10.86·10^−3^ K^−1^	385.20	8170.42	0.9232

**Table 3 molecules-27-00031-t003:** Parameters of the model approximation of the TODGA adsorption isotherm from a solution in methanol.

Model		
**Langmuir** $q=\frac{q_{s}\cdot K\cdot C}{1+K\cdot C}$	**Bi-Langmuir** $q=\frac{q_{s1}\cdot K_{1}\cdot C}{1+K_{1}\cdot C}+\frac{q_{s2}\cdot K_{2}\cdot C}{1+K_{2}\cdot C}$	**Freundlich** q=a·C1b
*q_s_* = 0.21 mmol/g, *K* = 1.06·10^3^ L/mol, *R*^2^ = 0.8251.	*q_s1_* = 0.11 mmol/g, *K*_1_ = 5.29·10^3^ L/mol, *q_s2_* = 0.42 mmol/g, *K*_2_ = 45.3 L/mol, *R*^2^ = 0.9076.	*a* = 0.107, *b* = 2.97, *R*^2^ = 0.9060.

## Data Availability

Not applicable.

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
