# Peer review of "Adsorption of *N*,*N*,*N*′,*N*′-Tetraoctyl Diglycolamide on Hypercrosslinked Polysterene from a Supercritical Carbon Dioxide Medium"

_molecules, 2021, doi:10.3390/molecules27010031_

Round 1
Reviewer 1 Report
This is relatively a good manuscript. The authors consider the use of TODGA as an adsorbent in the carrier of HCP in supercritical CO2 medium. The experimental was well designed and all the results were provided in detail. This paper is informative and clearly presented. It can be considered for publication in its present from, and the work is useful for readers with similar research background. The reviewer has no additional questions or comments.
additional commentary: The paper is really good and informative. The conclusion is too long, it should be rewritten to be more concise.
Author Response
Thank you for your review. We have edited our conclusions to make them more concise and readable.
Reviewer 2 Report
Although authors have taken up a informative topic for their work, however, there are few points that needs to be adressed
- The purpose of the study should be mentioned clearly, is the adsorbate toxic materials or it is wasted in abundant amount. The toxicity of the adsorbate material should be mentioned in the introduction part.
- How abundantly TOGDA with supercritical CO2 is available, how does is affect the environment?
- Authors should also mention the problems encountered during the experiments related adsorption of TOGDA, if any.
- A comparison of the dynamic and the static adsorption procedures from the reported literature should be mentioned at the appropriate place.
- For sorbent washing/ conditioning, did authors optimize the solvent and the washing time? Washing time has any role in adsorption?
- Did authors develop their own method for the quantitative analysis of TODGA? Any new method or any modification in the existing analytical method must be validated to make sure the results are accurate.
- Since the adsorption studies involves a lot of samples, what is the advantage of using HPLC instead of using fast technique say Spectrophotometry.
- Figure 3 what does peak 1 and peak 2 represent in the chromatogram, how can the same analyte appear at the different retention times?
- Corresponding Linear regression should be mentioned (figure 4)
- More detailed Gravimetric analysis needs to be mentioned in line 221-232.
- Figure 4 has limited significance; in my opinion it should be deleted.
- Autoclave filling with CO2 should be explained further.
- A comparative study of different kind of adsorbent used for the adsorption of N,N,N’,N’-tetraoctyldiglycolamide should be summarized in tabular form.
- Characterization of the adsorbent should be mentioned in detail.
Reviewer 3 Report
Please find my comments in the attached file.

Round 2
Reviewer 2 Report
Accept